# Generation of Current Good Manufacturing Practices-Grade Mesenchymal Stromal Cell-Derived Extracellular Vesicles Using Automated Bioreactors

**DOI:** 10.3390/biology14030313

**Published:** 2025-03-20

**Authors:** Elani F. Wiest, Abba C. Zubair

**Affiliations:** 1Department of Regenerative Biotherapeutics, Mayo Clinic, Jacksonville, FL 32224, USA; wiest.elani@mayo.edu; 2Department of Laboratory Medicine and Pathology, Mayo Clinic, Jacksonville, FL 32224, USA

**Keywords:** mesenchymal stromal cells, current good manufacturing practices, extracellular vesicles, bioreactors

## Abstract

Extracellular vesicles (EVs) are tiny particles released by cells that help in cell communication and maintaining balance in the body. They can be sourced from various body fluids and tissues, including blood, cerebrospinal fluid, and mesenchymal stromal cells (MSCs). MSCs are particularly favored for EV production due to their availability, ease of isolation, and ability to grow in culture. Manufacturing clinical-grade EVs requires following strict standards (cGMP). Using automated 3D bioreactors for cell culture is more efficient and scalable than traditional 2D methods. These bioreactors improve EV yield and quality, making the process more economical and consistent. Different cell culture systems, like 2D flasks and 3D bioreactors, have their pros and cons. Three-dimensional systems are generally better for large-scale production and maintaining cell characteristics. Techniques like tangential flow filtration (TFF) are preferred for isolating EVs due to their efficiency and scalability. Ensuring the purity and consistency of EVs is crucial. This involves validating every step of the manufacturing process and carefully controlling culture conditions. The use of serum in the culture media is debated, as it can affect EV composition. Overall, the field of EV manufacturing is still developing, and standardizing processes is essential for producing high-quality, clinical-grade EVs.

## 1. Introduction

Extracellular vesicles (EVs) are nanosized vesicles secreted by cells, which are bound by a lipid bilayer and contain no nucleus [1]. EVs play a pivotal role in normal physiology and pathophysiological states to restore homeostasis through cell–cell communication and are packaged with bioactive content like nucleic acids, metabolites, proteins, and lipids [2].

cGMP-grade EVs can be obtained from many sources, for example, plasma, cerebrospinal fluid (CSF), the stromal vascular fraction (SVF) of adipose tissue, and MSCs. Li et al. showed that a diverse source of EVs is present in healthy plasma samples. EVs from blood cells (B cells, T cells, RBCs, neutrophils, platelets, and monocytes) accounted for 99.8% of the total EV population [3]. Similarly, multiple studies have identified at least six different cell types in SVF, including MSCs and hematopoietic stem and progenitor cells [4,5,6,7], while CSF is rich in white blood cells [8]. It is yet to be determined which cell type in SVF and CSF contributes to the most EVs in each preparation.

MSCs gained overwhelming interest due to their safety profile [9,10]. Isolating EVs specifically from MSCs has many benefits. MSCs are readily available, easily isolated, easy to expand, and homogeneous in cell culture. MSCs can be isolated from various tissue sources, of which the most popular are bone marrow and adipose tissue [11]. MSCs are easy to identify according to the minimal criteria for defining multipotent mesenchymal stromal cells by the International Society for Cellular Therapy. The updated characterization criteria for MSCs include (1) a description of the tissue of origin, (2) a detailed description of positive and negative markers, and (3) in vitro functionality testing [12]. Specifically, many mesenchymal stromal cell (MSC)-derived EVs (MSC-EVs) have been shown to contain anti-inflammatory cytokines and growth factors like transforming growth factor beta 1 (TGFβ1), Interleukin-10 (IL-10), and hepatocyte growth factor (HGF) [13,14]. MSC-EVs are being investigated for the treatment of psoriasis [15], rotary cuff repair [16], wound healing [17], and other conditions in pre-clinical trials and early-phase clinical trials. It is essential to manufacture clinical-grade EVs according to cGMP standards. Manufacturing cost and batch-to-batch variability can be limited if all cells needed for a given study can be manufactured in one or two runs instead of six to seven runs. Therefore, the more EVs harvested per run, the less variability and need to compare to other runs.

A cGMP-compliant, economical, and scalable EV manufacturing platform is thus critical to support clinical trials. This review will discuss factors to consider when manufacturing cGMP-grade MSC-EVs from conditioned media in automated bioreactors.

## 2. Cell Culture Bioreactors for Manufacturing cGMP-Grade MSC-Derived EVs

The most popular system used to culture cells is a manual 2D cell culture flask system; however, other methods are also available. Cells cultured in growth media (usually supplemented with serum and L-glutamine) in a monolayer on rigid plastic flasks or Petri dishes is a century-old example of a 2D cell culture system. It is still popular today because it is relatively inexpensive and easy to use and maintain [18]. Three-dimensional cell culture systems have gained popularity as the need for more physiologically relevant and scalable in vivo models has grown. An in-depth review of 3D flask cell culture systems was published by Breslin et al. [18], and a thorough analysis of the physiological differences between 2D and 3D cultures was published by Duval et al. [19]. Comparative studies have demonstrated differences between 2D and 3D systems in terms of metabolic fingerprints [20], morphology, gene expression [21,22,23], exposure to nutrients and drugs, proliferation [24], and drug sensitivity [22]. Not surprisingly, gene expression of EVs from cells cultured in 2D vs. 3D systems has also been demonstrated to be different [23]. miRNA, EV markers, and protein cargo between 2D and 3D models differed in signature [25,26]. Additionally, the number of EV-derived miRNAs is increased when grown in 3D cell culture vs. 2D culture [25].

Our group has previously published a study comparing manufacturing MSCs via a hollow fiber bioreactor vs. cell stacks. We found that the different platforms did not impact release criteria regarding sterility, viability, and flow cytometry analyses. Furthermore, we found that the cost per dose was USD 979.41 less when using the bioreactor, hands-on manufacturing time was decreased by 326.6 h, and the cost to manufacture 100 doses was USD 97,942.47 lower when utilizing the bioreactor [27]. Using an automated 3D closed system to collect EVs has been demonstrated to be a more scalable, economical, and time-efficient method, and cell viability and EV harvest are also improved [27,28]. Another study showed that EVs manufactured in a 3D hollow fiber bioreactor are more efficacious. This was demonstrated by a decrease in anti-inflammatory markers and the tubular injury score in mice treated with 3D EVs compared to those treated with 2D EVs in a kidney injury mouse model [29].

Isolating conditioned media from 2D cell culture flasks requires opening the flask and replacing media manually every 2–3 days [30]. This open system creates multiple opportunities for contamination. Conditioned media must be centrifuged to remove large cell debris before isolating EVs using EV isolation techniques like TFF. If UC is used to isolate EVs, the additional centrifugation step can be omitted; however, UC has its own limitations: it is an open system, it introduces extra steps, it lacks scalability, it is time-consuming, and its footprint within a cleanroom environment is significant [31]. In addition, final product purity can be compromised with cell debris.

Closed-system 3D cell culture systems have an advantage in that they are automated, and conditioned media can be removed by sterile welding. The disadvantage of microbead-based bioreactors is that microbeads need to be removed from conditioned media, adding an extra step during which the conditioned media are sterile-filtered. Vacuum filtration is the only documented method to remove microbeads from conditioned media. Depending on the size of the filter, an additional centrifugation step may be required to remove other large debris [32,33]. This step adds further opportunities for contamination, and vacuum filtration can affect EV integrity and compromise scalability [34,35,36,37].

Hollow fiber 3D systems like the Quantum bioreactor generate liters of conditioned media during cGMP-grade MSC manufacturing. In one study, MSCs were seeded in the Quantum bioreactor at a density of 90–220 × 10^6^ MSCs with 250–500 mL of conditioned media being recirculated. On day 25, an average of 8.1 × 10^10^ particles/mL EVs were collected. In this study, MSCs were expanded in CellSTACK culture chambers before seeding in the Quantum [38]. Alternatively, conditioned media can be set to continuous flow to produce conditioned media continuously in a collection bag that can be removed aseptically. In another study, a 3D bioreactor with microbeads was used, and only 60 mL of conditioned media was collected for EV isolation, but it can be scaled to collect more conditioned media. Another caveat is that MSCs are grown to confluency, after which media are discarded, and only then are EV-containing conditioned media collected at a concentration of 4–7 × 10^9^ particles/mL [39]. The differences between EVs isolated from these overly confluent MSCs may differ from those collected from non-confluent MSCs. In many reports, MSCs are expanded in 2D cell culture flasks before seeding in 3D systems, adding unnecessary additional steps, cost, and time [38].

cGMP-compliant closed-system automated cell culture models have been on the market to generate not only MSCs and other cells but also EV-rich conditioned media. Table 1 summarizes the manufacturing pros and cons of the most widely used bioreactors.

## 3. Factors Affecting cGMP EV Manufacturing from MSC Cultures

cGMP-grade EVs can be obtained from MSCs grown in various automated and manual cell culture systems. Two-dimensional multilayer flasks can overcome some of the scalability issues of monolayer flasks; however, this is limited to incubator space, and harvesting cells in multilayer flasks is challenging, labor-intensive, and time-consuming [40]. As mentioned earlier, cells cultured in 2D do not necessarily translate to in vivo physiology [41,42]. While 3D cell culture systems are generally more physiologically relevant and scalable, they should ideally also be in a closed system to meet cGMP requirements for sterility.

De Almeida Fuzeta and colleagues demonstrated a 5.7-fold increase in EV yield from MSCs grown in a microbead-based 3D system vs. static cell culture flasks [39]. Microbead-based 3D systems used to expand and harvest cells are well documented; however, steps to separate microbeads from MSCs are not well described [33,42]. Since EVs do not attach to these microbeads, no separation is needed.

Stirred tank bioreactors that make use of cell aggregates can also be used. MSC aggregates omit the need to remove xenobiotic microcarriers from conditioned media when manufacturing EVs exclusively [43]. However, an additional centrifugation step to remove aggregated cells/microcarriers is still warranted. Another limitation of MSC aggregates is scalability due to limited diffusion of nutrients and gasses when aggregates increase in size. This can decrease the growth rate and EV yield if the aggregate size is not closely monitored [32,44].

Others have demonstrated that EV yield was significantly improved when MSCs were grown in stirred tank bioreactors. When seeding cells in a bioreactor and isolating them with tangential flow filtration (TFF), Haraszti showed a 140-fold increase in EV yield compared to those grown in tissue culture flasks and isolated by ultracentrifugation (UC). They further demonstrated a 7-fold yield increase compared to those grown in bioreactors and isolated by TFF [26].

Collecting cGMP-grade EVs from hollow fiber bioreactor systems has an advantage over stirred tank bioreactors [28,38]. Conditioned media can be collected aseptically by sterile welding the bag onto the bioreactor for EV isolation. The collected conditioned media do not require centrifugation since cell debris is limited in this fraction. This is beneficial when isolating EVs by TFF since TFF filters can be clogged by cells and other large debris.

## 4. Isolating cGMP-Grade EVs from Conditioned Media

### 4.1. cGMP-Grade EV Isolation Techniques

UC and TFF are the most popular cGMP-grade EV isolation techniques reported in the literature. UC is used to pellet small particles based on density and size. Large particles like cell debris and apoptotic bodies are precipitated at low speeds, while smaller particles like lipoproteins and EVs are pelleted at higher speeds (100,000 to 200,000× *g* range) [31,45,46]. Therefore, the EV fraction can contaminate other particles with similar density and size profiles. Washing steps are needed to purify the EV pellet, which can significantly affect the EV yield. In addition, EVs are often subject to vacuum filtration to remove large debris before UC, which hinders the integrity of EVs [34,35,36,37].

On the other hand, TFF is scalable, delivers EVs of high purity, can be conducted in a closed system, and consumes less footprint in a cleanroom environment than UC. TFF is a filtration process where the fluid flows parallel to the filter and is pushed through the pores based on size. This limits clogging and damage to filter pores and results in a more time-efficient, high-yield process. In addition, the filtrate undergoes diafiltration, during which cell culture media can be replaced by a cryoprotectant and concentrated to the desired level. Depending on the application, multiple TFF filters with different pore sizes can be used. TFF can also be paired with another method such as size exclusion chromatography; however, this usually is not cGMP-compliant or lacks scalability [26,47,48].

EVs can be isolated using the asymmetric-flow field-flow fractionation (AF4) technology. This technology leverages a combination of channel flow and crossflow to separate EVs based on their hydrodynamic size, ensuring high-resolution separation and purity. GMP-ready size exclusion chromatography columns are also available on the market; however, many lack scalability and consist of multiple open steps, increasing the risk of contamination [49].

### 4.2. Scalability

Scalability is a requirement for cGMP-grade EVs [50]. In addition to isolating EVs from conditioned media, generating a scalable volume of conditioned media is critical for manufacturing cGMP-grade EVs.

Conditioned media obtained from 2D cell culture are limited to the number of flasks used to culture MSCs. Collected conditioned media should be optimized for storage to isolate EVs over a few days simultaneously. Storage conditions for collected conditioned media should be investigated, and a scalable EV isolation technique should be used [51].

### 4.3. Process Validation

Process validation of every manufacturing step is necessary [52]. Inter-donor validations should be performed with every batch of MSCs, and several other validations must be performed. The first validation is to evaluate EV batches collected on different days. Vast differences have been observed in EV markers and concentration based on the day of collection from the Quantum 3D system [38].

Our group previously published a novel flow cytometry-based technique to assess batch-to-batch consistency [53]. Rohde et al. published a review discussing in-depth characterization and batch-to-batch consistency [52]. Furthermore, the MISEV guidelines discuss multiple techniques used and optimized for validation purposes [54].

Sample validation steps for cGMP manufacturing of EVs are outlined in red in Figure 1. The first validation pertains to donor screening and master bank generation. The second validation pertains to the EV-source cells, followed by EV product validation and post-thaw validation. Every validation step should assess identity, purity, reproducibility, sterility, safety, and potency [50].

Due to the heterogeneous nature of EVs, it is difficult to identify a specific mechanism of action, and therefore, many others have adopted the “the process is the product” approach [55]. This approach focuses on batch-to-batch consistency.

### 4.4. Serum Usage in EV Manufacturing

The use of any serum in EV manufacturing is highly debated. It depends on the clinical application and whether MSCs are manufactured simultaneously, since serum starvation may affect MSCs and EV cargo. If human platelet lysate is used, validations should be performed to determine the percentage of serum EVs in the product, and protocols should be developed to limit the number of serum EVs, for example, using EV-depleted serum or chemically defined media that lack serum components. Potency assays and EV characterizations should be validated for serum-free and serum-containing conditions, similar to manipulating other culture conditions, such as hypoxia or confluency. In one study, human umbilical cord MSCs (UC-MSCs) were cultured in media containing 10% FBS vs. chemically defined media. The authors found that chemically defined media improved wound healing by upregulating growth factors EGF and PDGF-AB/BB, while downregulating pro-inflammatory cytokines (IL-6 and IL-8) [17]. Similarly, Dao et al. investigated differences between FBS-supplemented, human platelet lysate (hPL)-supplemented, and three different chemically defined sources of media. They found that hPL-supplemented media promoted wound healing and exhibited increased concentrations of growth factors [56].

A purity assay validation is also warranted to ensure that the isolation technique removes all cell debris, microcarriers, and other contaminants in the conditioned media [48]. EV characterizations include identification (e.g., Western blot analysis and flow cytometry), size analysis (e.g., Nanosight Tracking Analysis and dynamic light scattering), and morphology (e.g., transmission electron microscopy and atomic force microscopy) [1,57].

## 5. Conclusions

The EV field is still in its infancy. The MISEV guidelines provide some guidance; however, there are no set standards for manufacturing cGMP-grade EVs. We previously discussed the challenges and best approaches for developing GMP-grade EVs [47]. Bioreactors can be utilized to manufacture cGMP-grade EVs in a more scalable and economical manner that is less prone to technical errors. Many published studies manipulate cell culture conditions to increase the number of EVs or serum-starve MSCs to eliminate serum-derived EVs. These alterations in cell culturing protocols may address the scalability and xenobiotic concerns. However, a change in cell culture conditions also induces a change in the EV population. This illustrates the importance of identifying an application and characterizing the EVs in detail. This includes validation of all steps, appropriate release testing, and assuring batch-to-batch consistency for all runs. Figure 1 summarizes the considerations and steps for the automated manufacturing of cGMP-grade EVs.

## Figures and Tables

**Figure 1 biology-14-00313-f001:**
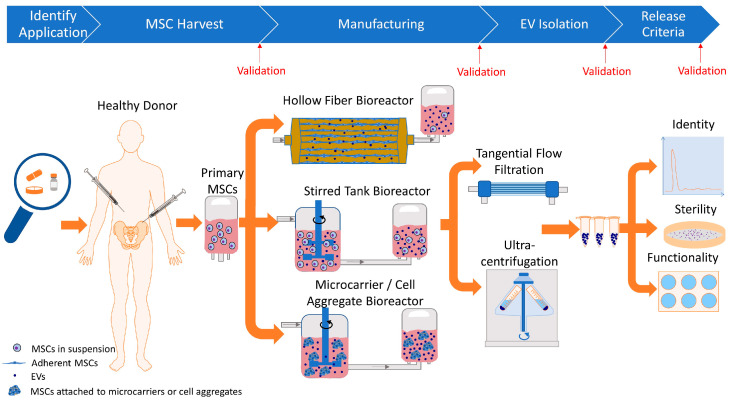
Considerations and steps for automated manufacturing of cGMP-grade EVs from conditioned media. After an application has been identified, a healthy donor is selected, and primary MSCs are harvested. MSCs are cultured in automatic bioreactors. Conditioned media are collected, and EVs are isolated from the conditioned media. The final step is confirming that the product meets all release criteria and is safe to be released for clinical application. Validation runs are performed after every step, as indicated in red. Figure created with Motifolio Toolkits (Motifolio Inc, Ellicott City, MD, USA).

**Table 1 biology-14-00313-t001:** Summary of pros and cons of different types of bioreactors to manufacture cGMP-grade EVs.

Bioreactor Type	Pros	Cons
**Hollow Fiber Bioreactors**	Scalable High yield due to large surface area for cell growth Closed system Protocol can be optimized to distinguish between media-derived and cell-derived EVs Continuous media exchange	Cell growth cannot be monitored accurately in-process Large initial upfront cost
**Stirred Tank Bioreactors**	Scalable Closed system (most) Continuous aeration Direct monitoring of cell growth in-process	Extra steps needed to separate cells and EVs High shear forces may impact cell viability and EV quality Extra media optimization may be needed to eliminate media-derived EVs
**Microcarrier Bioreactors**	Scalable Closed system (most) Increased surface area for cell growth Direct monitoring of cell growth in-process	Extra steps needed to separate cells and EVs Additional costs associated with GMP-grade microcarriers High shear forces may impact cell viability and EV quality Clumping may affect nutrient access Extra media optimization may be needed to eliminate media-derived EVs

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
