# Peer review of "Generation of Current Good Manufacturing Practices-Grade Mesenchymal Stromal Cell-Derived Extracellular Vesicles Using Automated Bioreactors"

_biology, 2025, doi:10.3390/biology14030313_

Round 1
Reviewer 1 Report
Comments and Suggestions for Authors
The article is a solid general overview of MSC-derived EV production in bioreactors, but it falls short in critical discussion, regulatory aspects, and practical implementation. Strengthening these areas would significantly enhance its scientific value and impact.
10 Key Weaknesses of the Reviewed Article:
-
Lack of Originality – The article primarily summarizes existing work without offering new insights or innovative approaches. It does not introduce novel findings or strategies to improve EV biomanufacturing.
-
Insufficient Critical Discussion – While the article covers various bioreactor systems, it lacks a critical comparative analysis of their strengths and weaknesses. A deeper discussion on how different bioreactors influence EV composition and quality would make the review more robust.
-
Overgeneralization of cGMP Requirements – The discussion on cGMP compliance remains vague, lacking concrete steps or case studies illustrating successful implementation. It does not address specific regulatory hurdles or detailed quality control measures.
-
Limited Coverage of Downstream Processing – Although tangential flow filtration (TFF) and ultracentrifugation (UC) are mentioned, newer EV purification technologies, such as size exclusion chromatography (SEC) or asymmetrical flow field-flow fractionation (AF4), are not properly discussed.
-
Minimal Focus on EV Characterization – The article does not provide a comprehensive evaluation of EV characterization techniques, such as proteomic and lipidomic analysis, which are essential for batch consistency in clinical applications.
-
Insufficient Quantitative Data on Bioreactor Optimization – While 3D bioreactors are emphasized, the article lacks quantitative comparisons on factors like cell viability, doubling time, metabolic activity, and EV productivity across different bioreactor systems.
-
Limited Clinical Relevance – The review does not explore preclinical and clinical evidence supporting the efficacy of MSC-derived EVs for specific therapeutic applications. A more detailed discussion on the mechanism of action of EVs in regenerative medicine would enhance its impact.
-
Oversimplified Discussion on Serum-Free Media – The article touches on the debate between serum-free and serum-containing culture conditions but does not present a direct experimental comparison or explore chemically defined media as an alternative.
-
Lack of Standardization Discussion – While acknowledging the lack of standardization in EV production, the article fails to suggest specific solutions, guidelines, or benchmarks for standardizing yield, purity, and potency in large-scale manufacturing.
-
Superficial Cost-Benefit Analysis – The financial viability of 3D bioreactors versus 2D systems is mentioned but not analyzed in depth. Including factors such as cost per batch, labor requirements, scalability, and production efficiency would improve the review’s practical value.
Author Response
The article is a solid general overview of MSC-derived EV production in bioreactors, but it falls short in critical discussion, regulatory aspects, and practical implementation. Strengthening these areas would significantly enhance its scientific value and impact.
10 Key Weaknesses of the Reviewed Article:
1. Lack of Originality – The article primarily summarizes existing work without offering new insights or innovative approaches. It does not introduce novel findings or strategies to improve EV biomanufacturing.
This review article aims to address the translation from basic research to cGMP-grade EVs. Many bioreactor approaches have historically been implemented in manufacturing MSCs and other cells. These approaches need to be adapted to be able to manufacture EVs, which are significantly smaller than whole cells and face other challenges like contamination from serum that needs to be considered.
2. Insufficient Critical Discussion – While the article covers various bioreactor systems, it lacks a critical comparative analysis of their strengths and weaknesses. A deeper discussion on how different bioreactors influence EV composition and quality would make the review more robust.
The authors agree that more data is needed to directly compare EV composition and quality between different bioreactors. Upstream process development studies should identify the best bioreactor and release criteria should be designed to critically assess batch-to-batch consistency. The authors added Table 1 to summarize general pros and cons between different bioreactors.
3. Overgeneralization of cGMP Requirements – The discussion on cGMP compliance remains vague, lacking concrete steps or case studies illustrating successful implementation. It does not address specific regulatory hurdles or detailed quality control measures.
There is currently no official regulatory guidance for EVs and data on specific release criteria used in EVs currently in clinical trials are not available. The authors cited a previous manuscript (ref 47) to discuss the general cGMP requirements as outlined by the FDA for cell products.
4.Limited Coverage of Downstream Processing – Although tangential flow filtration (TFF) and ultracentrifugation (UC) are mentioned, newer EV purification technologies, such as size exclusion chromatography (SEC) or asymmetrical flow field-flow fractionation (AF4), are not properly discussed.
Although the discussion of isolation methods is beyond the scope of this review, the authors added the following paragraph to section 4.2: “EVs can be isolated using the asymmetric-flow field-flow fractionation (AF4) technology. This technology leverages a combination of channel flow and crossflow to separate EVs based on their hydrodynamic size, ensuring high-resolution separation and purity. GMP-ready size exclusion chromatography columns are also available on the market, however many lack scalability and consist of multiple open steps, increasing the risk of contamination.”
5. Minimal Focus on EV Characterization – The article does not provide a comprehensive evaluation of EV characterization techniques, such as proteomic and lipidomic analysis, which are essential for batch consistency in clinical applications. EV Characterization is beyond the scope of this manuscript, however the authors added the following to section 4.5: “Our group previously published a novel flow cytometry-based technique to assess batch-to-batch consistency (Korchak et al., 2023). Rohde et al. published a review discussing in-depth characterization and batch-to-batch consistency (Rohde, 2019). Furthermore, although the MISEV guidelines discuss multiple techniques used and optimized for validation purposes (Welsh et al, 2023).”
6. Insufficient Quantitative Data on Bioreactor Optimization – While 3D bioreactors are emphasized, the article lacks quantitative comparisons on factors like cell viability, doubling time, metabolic activity, and EV productivity across different bioreactor systems.
The authors agree that there is no primary literature comparing EV characteristics between different bioreactors.
7. Limited Clinical Relevance – The review does not explore preclinical and clinical evidence supporting the efficacy of MSC-derived EVs for specific therapeutic applications. A more detailed discussion on the mechanism of action of EVs in regenerative medicine would enhance its impact.
The following was added to the introduction section: “MSC-EVs are being investigated for the treatment of psoriasis (Lai et al., 2023), rotary cuff repair (Jenner et al., 2023), wound healing (Kim et al., 2023), and other conditions in pre-clinical trials and early-phase clinical trials.” Going in depth to discuss the mechanism of action of EVs in regenerative medicine is beyond the scope this manuscript.
8. Oversimplified Discussion on Serum-Free Media – The article touches on the debate between serum-free and serum-containing culture conditions but does not present a direct experimental comparison or explore chemically defined media as an alternative.
The following was added to the serum section: “In one study, human umbilical cord MSCs (UC-MSCs) were cultured in media containing 10% FBS vs. chemically defined media. The authors found that chemically defined media improved wound healing by upregulating growth factors EGF and PDGF-AB/BB, while downregulating pro-inflammatory cytokines (IL-6 and IL-8) (Kim et al., 2021). Similarly, Dao et al., investigated differences between FBS-supplemented, human platelet lysate (hPL)-supplemented, and three different chemically defined media sources. They found that hPL-supplemented media promoted wound healing and exhibited increased concentrations of growth factors. (Dao et al., 2024).”
9. Lack of Standardization Discussion – While acknowledging the lack of standardization in EV production, the article fails to suggest specific solutions, guidelines, or benchmarks for standardizing yield, purity, and potency in large-scale manufacturing.
The authors added the following sentence and citation to the conclusion section: “We previously discussed the challenges and best approaches for developing GMP-grade EVs [47].”
10. Superficial Cost-Benefit Analysis – The financial viability of 3D bioreactors versus 2D systems is mentioned but not analyzed in depth. Including factors such as cost per batch, labor requirements, scalability, and production efficiency would improve the review’s practical value.
The authors expanded on the reference cited in section 2: “Our group has previously published a study comparing manufacturing MSCs via a hollow-fiber bioreactor vs. cell stacks. We found that the different platforms did not impact release criteria regarding sterility, viability, and flow cytometry analyses. Furthermore, we found that the cost per dose was $979.41 less when using the bioreactor, hands-on manufacturing time was decreased by 326.6 hours, and the cost to manufacture 100 doses was $97,942.47 lower when using the bioreactor [25].”
Reviewer 2 Report
Comments and Suggestions for Authors
I have reviewed the article Generation of cGMP-grade MSC Derived Extracellular Vesicles Using Automated Bioreactors and recommend its acceptance by the journal. It is a well-structured and clearly organized review that, despite its concise approach, presents relevant information in a precise and well-supported manner. The content is well-developed and effectively fulfills its purpose. However, I do not feel qualified to assess the quality of the English writing.
Author Response
1. I have reviewed the article Generation of cGMP-grade MSC Derived Extracellular Vesicles Using Automated Bioreactors and recommend its acceptance by the journal. It is a well-structured and clearly organized review that, despite its concise approach, presents relevant information in a precise and well-supported manner. The content is well-developed and effectively fulfills its purpose. However, I do not feel qualified to assess the quality of the English writing.
The authors thank the reviewer for the positive feedback.
Reviewer 3 Report
Comments and Suggestions for Authors
The manuscript reports on the modalities of obtaining clinical grade extra-cellular vescicles using authomated bioreactors.
The paper is well written and provides an overview of the modalities through which one can obtain extra-cellular vescicles to be used for diagnostic and therapeutic purposes.
The manuscript is a modest overview of what is known and reported in the field.
Additional comments: As I said, the manuscript is lacking rigor and solidity, being only an evaluation of works done by other authors. Furthermore, the submitting authors did not have a solid record of publications in the field.
I considered it useless to add any specific details, being the manuscript not worth accepting.
Author Response
1. The manuscript reports on the modalities of obtaining clinical grade extra-cellular vescicles using authomated bioreactors.The paper is well written and provides an overview of the modalities through which one can obtain extra-cellular vescicles to be used for diagnostic and therapeutic purposes.The manuscript is a modest overview of what is known and reported in the field.Additional comments: As I said, the manuscript is lacking rigor and solidity, being only an evaluation of works done by other authors. Furthermore, the submitting authors did not have a solid record of publications in the field.
I considered it useless to add any specific details, being the manuscript not worth accepting.
Response: Our review focuses on the emerging field of extracellular vesicles (EV), specifically clinical trials involving GMP-grade EVs. There is limited expertise in generating cGMP-grade MSC-derived EV using automated bioreactors. To our knowledge, this is the first review on this topic. We believe that our manuscript has significantly improved with the addition of our response to the reviewers' comments, making it worthy of sharing with your journal audience.
Reviewer 4 Report
Comments and Suggestions for Authors
The review is devoted to a relevant topic, but many issues are not covered, and the statements remain unsubstantiated and require reading the cited literature for understanding. The introduction should explain cGMP-grade EVs, what requirements are currently imposed on EVs and what quality criteria are used. In the definition of MSC it is necessary to explain the concept of "in vitro functionality". "MSCs can be passaged multiple times in vitro, differentiating them from other hematopoietic cells" - since MSCs are not hematopoietic cells, the sentence needs to be rephrased. Section 2 describes the advantages of 3D cell culture vs. 2D culture, but what are the disadvantages? And in listing the works that showed the differences between them, it is desirable to specify and generalize these differences. Why MSC aggregates omit the need to remove xenobiotic microcarriers from conditioned media? In Section 3, presenting the pros and cons of different methods in a table will make things much easier to understand. Most of the text in sections 4.1 and 4.2 belongs more to section 2. Section 4.5 needs to explain what exactly needs validation and at what manufacturing stage, otherwise the statement that validation is important is lost.
Author Response
1. The review is devoted to a relevant topic, but many issues are not covered, and the statements remain unsubstantiated and require reading the cited literature for understanding. The introduction should explain cGMP-grade EVs, what requirements are currently imposed on EVs and what quality criteria are used. In the definition of MSC it is necessary to explain the concept of "in vitro functionality". "MSCs can be passaged multiple times in vitro, differentiating them from other hematopoietic cells" - since MSCs are not hematopoietic cells, the sentence needs to be rephrased. Section 2 describes the advantages of 3D cell culture vs. 2D culture, but what are the disadvantages? And in listing the works that showed the differences between them, it is desirable to specify and generalize these differences. Why MSC aggregates omit the need to remove xenobiotic microcarriers from conditioned media? In Section 3, presenting the pros and cons of different methods in a table will make things much easier to understand. Most of the text in sections 4.1 and 4.2 belongs more to section 2. Section 4.5 needs to explain what exactly needs validation and at what manufacturing stage, otherwise the statement that validation is important is lost.
Response: The authors would like to thank the reviewer for the constructive comments. The manuscript has undergone editing to expand on cited studies and validation steps. We have also generated Table 1 to illustrate the pros and cons of the discussed bioreactors. The suggested sections have been reorganized and shifted, and the unclear sentences have been rewritten.
Round 2
Reviewer 3 Report
Comments and Suggestions for Authors
The authors replied to the majority of issues raised.
Reviewer 4 Report
Comments and Suggestions for Authors
The authors made the modifications and clarifications requested during the review.